# Selective Angiographic Flat Detector Computer Tomography Blood Volume Imaging in Pre-Operative Vascular Mapping and Embolization of Hypervascular Intracranial Tumors—Preliminary Clinical Experience

**DOI:** 10.3390/diagnostics12051185

**Published:** 2022-05-10

**Authors:** Thijs van der Zijden, Annelies Mondelaers, Caro Franck, Maurits Voormolen, Tomas Menovsky

**Affiliations:** 1Department of Radiology, Antwerp University Hospital (UZA), 2650 Edegem, Belgium; caro.franck@uza.be (C.F.); maurits.voormolen@uza.be (M.V.); 2Department of Medical Imaging, AZ Klina, 2930 Brasschaat, Belgium; 3Research Group mVision, Faculty of Medicine and Health Sciences, University of Antwerp (UA), 2610 Antwerp, Belgium; 4Department of Neurosurgery, Antwerp University Hospital (UZA), 2650 Edegem, Belgium; tomas.menovsky@uza.be; 5Research Group Translational Neurosciences, Faculty of Medicine and Health Sciences, University of Antwerp (UA), 2610 Antwerp, Belgium

**Keywords:** angiography, embolization, hypervascular tumors, parenchymal blood volume, cone-beam computed tomography, perfusion imaging

## Abstract

Pre-operative embolization of hypervascular intracranial tumors can be performed to reduce bleeding complications during resection. Accurate vascular mapping of the tumor is necessary for both the correct indication setting for embolization and for the evaluation of the performed embolization. We prospectively examined the role of whole brain and selective parenchymal blood volume (PBV) flat detector computer tomography perfusion (FD CTP) imaging in pre-operative angiographic mapping and embolization of patients with hypervascular intracranial tumors. Whole brain FD CTP imaging with a contrast injection from the aortic root and selective contrast injection in the dural feeding arteries was performed in five patients referred for tumor resection. Regional relative PBV values were obtained pre- and post-embolization. Total tumor volumes with selective external carotid artery (ECA) supply volumes and post-embolization devascularized tumor volumes were determined as well. In all patients, including four females and one male, with a mean age of 54.2 years (range 44–64 years), the PBV scans were performed without adverse events. The average ECA supply was 54% (range 31.5–91%). The mean embolized tumor volume was 56.5% (range 25–94%). Relative PBV values decreased from 5.75 ± 1.55 before embolization to 2.43 ± 1.70 post-embolization. In one patient, embolization was not performed because of being considered not beneficial for the resection. Angiographic FD CTP imaging of the brain tumor allows 3D identification and quantification of individual tumor feeder arteries. Furthermore, the technique enables monitoring of the efficacy of pre-operative endovascular tumor embolization.

## 1. Introduction

Surgical resection of hypervascular, cranial tumors, such as meningiomas, hemangiopericytoma and hemangioblastomas, can potentially pose significant intra-operative problems due to excessive bleeding. In lowering the bleeding risk during surgery, pre-operative catheter angiography with subsequent preventive embolization of the tumor can be considered [1,2,3,4]. Extra-axial tumors may receive an arterial blood supply from dural, pial or mixed dural-pial arteries. In practice, only dural arterial feeders are considered suitable candidates for preventive, pre-operative embolization. Complete devascularization is often difficult to achieve. Furthermore, embolization procedures are not without risks and potentially devastating neurological complications after embolization can occur due to unintentional embolization of eloquent tissue, e.g., the eye, cranial nerves or normal brain tissue, or tumor swelling [5,6,7]. Rosen et al. reported permanent neurological deficits in 9% of 167 patients with cranial base meningiomas undergoing pre-operative embolization [8]. With the help of careful angiographic mapping of the vascular supply of the tumor, embolization might be performed in a relatively safe way and a significant portion of the tumor can be embolized [9]. Embolization can be achieved by injecting microparticles and/or glue by means of a microcatheter in a tumor feeder [3]. In standard clinical practice, indication settings for pre-operative embolization and evaluation of performed embolotherapy are both based on an assessment of the degree of tumor contrast staining on conventional 2D digital subtraction angiography (DSA) images. However, it can be difficult to identify proper candidates for embolization and to assess the efficacy of the intervention based on 2D DSA alone. Therefore, pre- and post-procedural conventional cross-sectional imaging (e.g., computer tomography (CT) and magnetic resonance imaging (MRI)) can be helpful in providing additional structural information about the vascular tumor for optimal planning of the embolization and evaluation of the embolization result [10]. Nowadays, it is possible to perform flat detector CT (FD CT) imaging in the angiography suite [11]. This modality gives the interventionalist on-the-spot insights into parenchymal anatomy, vascular anatomy and perfusion of tumors, enabling more efficient decision-making during angiography and embolization procedures without the need for transporting patients from the angiography suite. Several studies underlined the pivotal role of FD CT techniques in interventional neuroradiology practice [12,13,14,15,16]. Current high-end angiographic systems can carry out perfusion imaging with dynamic perfusion parameters (e.g., cerebral blood flow (CBF), contrast transit time parameters, time to peak, etc.) [17]. However, in most clinical interventional neuroradiology practices, angiographic perfusion imaging is limited to a static parameter during the steady-state contrast injection phase due to the limitation of C-arm rotations in short-time intervals, rendering so-called parenchymal or pooled blood volume (PBV) maps. The acquired PBV values seem to correspond to the cerebral blood volume (CBV) values obtained with conventional CT [18] and MRI [19,20]. In contrast to conventional cross-sectional perfusion imaging techniques, angiographic FD CT perfusion (FD CTP) examination is not only able to image the entire brain, but also has the potential to visualize exclusive perfusion areas by means of acquisition during selective contrast injection in feeding arteries. This allows a very precise 3D delineation of the feeder supply area and volume compared to the total tumor volume. In this paper, we demonstrate the clinical application of the angiographic FD CTP PBV imaging technique and explore its utility for precise identification and quantification of individual tumor feeder arteries and determination of pre-operative tumor embolization efficacy.

## 2. Materials and Methods

### 2.1. Patients

The hospital’s local ethics committee (full name: “Comité voor Medische Ethiek UZA-UA”) approved the study protocol on 28 May 2018 (ref. no. B300201836599). Written informed consent from all patients was obtained prior to inclusion.

A total of 5 patients (1 male and 4 female), aged between 44 and 64 years (mean age 54.2), referred for resection of hypervascular cranial tumors by the neurosurgery team of our institution, were included in our prospective study between February 2018 and November 2020. The potential benefit for pre-operative embolization of a tumor was based on pre-operative MRI examinations. The probable tumor diagnosis and localization were determined. A hypervascular nature of a tumor was suspected based on imaging findings, such as vivid tumor enhancement, the presence of associated large vessels and the presence of intralesional flow voids. As part of the clinical routine, a conventional workup with conventional angiographic and angiographic perfusion examinations was performed before the planned surgery. The exclusion criteria were the following: patients with renal impairment, allergy to iodinated contrast or other contraindications to DSA, and pregnancy. An overview of the tumor clinical features, including diagnosis based on a pathological examination of resection specimens, tumor location and arterial supply, is displayed in Table 1. Operating time (hours) and blood loss (mL) during surgical resection were documented as well. Pathological data revealed that all cases were meningioma (Table 1).

### 2.2. Angiographic Imaging

Angiographic examination, including conventional 2D DSA, was performed on a biplane angiographic system (Artis Zee with Pure^®^ biplane system, Siemens Healthcare GmbH, Forchheim, Germany). FD CTP PBV imaging acquisition was done by two rotations of 6 s each, in which 397 frames were obtained at a total scan angle of 200°. The first rotation consisted of a mask run without the contrast injection and the second run, the fill run, was performed after the contrast injection. To obtain FD CTP of the whole brain, a contrast medium was injected through a 4 French pigtail catheter, positioned at the level of the aortic root. A total of 75 mL, consisting of a mixture of 25 mL contrast medium (Iomeron^®^ 300, Bracco, Milan, Italy) with 50 mL saline, was injected at a rate of 5 mL/s. Nine seconds after the start of contrast injection, the fill run was carried out. Selective contrast injection in tumor feeder arteries was done by a 4 French diagnostic catheter. An amount of 4 mL contrast medium (Iomeron^®^ 300, Bracco) mixed with 16 mL saline, was injected at a rate of 2 mL/s injection rate with a 2 s scan delay. Angiographic FD CTP PBV imaging was done during steady-state contrast quantity in the brain because the used C-arm is not suited for dynamic perfusion imaging. Selective feeder artery injection was performed only in dural feeding arteries from the external carotid artery circulation and before any embolization. After embolization, whole brain FD CTP was repeated.

The additional radiation dose exposure associated with the added FD CT PBV acquisitions was analyzed, using the archived dose reports of the angiographic examinations. The brain dose (BD) and effective dose (E) of individual acquisitions were calculated using the registered dose area products with the use of a Monte Carlo software tool (PCXMC version 2.0.1.4 Rotation, Radiation and Nuclear Safety Authority STUK, Helsinki, Finland).

### 2.3. Image Processing

After the acquisition, the color-coded PBV maps were calculated in units of mL contrast/1000 mL tissue using commercially available syngo DynaPBV Neuro software (Siemens Healthcare GmbH, Forchheim, Germany). We used a slice-wise relative PBV (rPBV) calculation method according to Wen et al. [21]: rPBV = 0.5 × (PBVtumor axial + PBVtumor sagittal)/PBVhealthy. The axial and sagittal PBV values were calculated using freehand regions of interest (ROI) surfaces at the largest tumor portions in the respective planes. The PBV maps were correlated with the fill run series for correct delineation of the tumor, using the fusion application tool on the angiographic workstation. In this way, we were able to define tumor margins more clearly. Healthy PBV values were obtained using circular ROIs in the sagittal plane in corresponding contralateral brain areas (Figure 1). The difference in rPBV pre- and post-embolization (∆rPBV) was measured as well. The mask runs were available for use as non-enhanced CT scan series. The fill runs were applied to render FD CT angiography images. The three obtained series (mask run, fill run and PBV maps) were post-processed on a Leonardo workstation to maximum intensity map (MIP) images in para-axial, para-sagittal and para-coronal views. Tumor feeding arteries were divided into the dural external carotid artery (ECA) and pial supply arteries. Tumor volumes acquired with aortic contrast injection, rendering whole brain perfusion imaging, and selective perfusion volumes, obtained during selective contrast injection in dural feeder arteries, i.e., external carotid arteries, were calculated using the fill run datasets for all patients in mL by use of syngo 3D Segmentation software. The volume of total tumor feeding artery supply areas, potentially eligible for embolization, were computed in percentages in relation to total tumor volume (Table 2).

### 2.4. Embolization Procedure

All embolization procedures were performed in patients under general anesthesia. After the angiographic workup, in consent with the operating neurosurgeon, suitable tumor feeder arteries were selected for embolization. Embolization was done by injecting microparticles (size ranging from 100–500 µm), Gelfoam fragments and/or non-adhesive liquid embolic agents (Onyx^®^, PHIL^®^) through selectively positioned micro-catheters. The choice of whether to use particles or liquid embolic agents was based on the discretion of the interventionalist. Liquid embolic agents were only used if the catheter tip was deemed close to or inside the tumor. Particles were used if the catheter tip was positioned more distally from the tumor. The endpoint of embolization was decided case by case by the interventionalist based on the judgment of the amount of tumor opacification at selective DSA imaging during the procedure, in accordance with the findings derived from post-embolization PBV imaging. Surgery was performed ranging from immediately after embolization on the same day until up to 2 days later.

### 2.5. Statistical Analysis

Besides the descriptive statistics, we attempted data analysis for the comparison of tumor volume and rPBV values before and after embolization by using the IBM SPSS software (version SPSS 27; SPSS Inc; Chicago, IL, United States of America) with the criterion for statistical significance defined as *p* ≤ 0.05.

## 3. Results

The angiographic PBV perfusion acquisitions were technically successful in all our patients. No adverse events occurred during the angiographic examinations or during or after the embolization procedures. In all patients, a mixed pial and dural vascular supply pattern was present. Figure 2 shows an example of the PBV images obtained after selective and aortic arch injections pre- and post-embolization, respectively.

In patient no. 3, with a transitional meningioma located at the left sphenoid ridge, a predominant pial vascular supply was present. No embolization was done because it was considered not helpful for the resection (Figure 3).

The operation time varied between 3 h and 40 min (patient no. 5) to 14 h (patient no. 3) of surgery. The longest operation time correlated with the patient with the predominant pial arterial supply (patient no. 3, Figure 3), in which an external to internal bypass (from superficial temporal artery to middle cerebral artery) was constructed. Negligible blood loss (<50 mL) was present in patients no. 2, 4 and 5. In patients no. 1 and 3, a blood loss of 500 mL and 300 mL was observed, respectively. The post-operative course was uneventful for all patients.

The calculated average brain dose and effective dose of one PBV acquisition were 36.3 mSv and 2.5 mSv, respectively.

In the four embolization patients, a mean difference of 23.5 ± 12.18 mL in devascularized tumor volume between pre- and post-embolization status was calculated (*p* = 0.125), which corresponded with an average percentage of devascularized tumor volume of 56.5%, ranging from 94% to 25%.

We observed a decrease in mean rPBV from 5.75 ± 1.55 before embolization to 2.43 ± 1.70 post-embolization (*p* = 0.125), resulting in a mean difference of 3.33 ± 2.04. The power to find an effect of this size with a paired t-test on four observations is 60%. The statistical power for the determination of a statistically significant effect was therefore too low in this sample, given the very small number of patients.

In patient no 2, we observed a lower ∆rPBV (0.7) compared with the ∆rPBV of the other embolized patients (patients no 1, 4 and 5 with ∆rPBV’s of 5.6, 3.9 and 3.1, respectively). Total external carotid artery dural vascular supply to the tumors varied from 31.5% to 91%.

## 4. Discussion

Limited publications exist about the clinical application of perfusion imaging techniques in the angio suite during angiographic and interventional procedures. Except for some recent high-end angiographic systems, commonly used angiographic systems are only able to acquire static 3D perfusion images, due to the lack of temporal resolution of the FD CT scan sweeps. The nature of FD CTP PBV values obtained during the steady-state contrast filling state is not entirely consistent. A recent systematic review of the literature, comparing the diagnostic accuracy of FD CTP with CT and MRI perfusion, stated that additional studies were required. Although, it has been demonstrated that FD CT provided similar CBV values and reconstructed blood volume maps as CT perfusion in cerebrovascular disease [22]. Struffert et al. showed that PBV values, obtained from a small patient group with acute ischemic stroke in the anterior circulation, corresponded less with the CBV values acquired with MRI [20]. Kamran et al. reported that in a series of 26 patients with delayed cerebral ischemia (DCI) after aneurysmal subarachnoid hemorrhage, the acquired PBV values corresponded best with a “CBF weighted” CBV parameter compared to MRI perfusion. Interestingly, besides comparing the FD CT PBV values to the MRI perfusion parameters, the authors demonstrated the potential benefit of using PBV perfusion in monitoring the treatment of patients with DCI [19].

Regarding the use of perfusion imaging in the evaluation of embolization efficacy of hypervascular tumors in the angio suite, a recent paper described the results of 18 patients with hypervascular brain tumors using the 6 s PBV technique before and after pre-operative tumor embolization [21]. Semi-quantitative analysis was performed to determine the feasibility of detecting perfusion deficits and the success of the embolization procedure. They reported a significant decrease in tumor rPBV values after embolization compared to the values before embolization. The calculated post-embolic tumor perfusion indices correlated significantly with blood losses during tumor resection and surgery times. They concluded that FD CT PBV imaging is a valuable method for evaluating hypervascular brain tumor perfusion and embolization efficacy. In contrast to our study, they only performed FD CTP acquisitions during injection of diluted contrast in the aortic arch by a pigtail catheter and did not obtain PBV images during selective tumor feeder injection. Our study used both selective and non-selective angiographic 3D FD CTP imaging to produce additional insights into both the tumor vascular supply mapping and pre-operative embolization results.

Martin et al. (2007) published a study about MRI perfusion imaging with selective tumor feeder contrast injection in the angiography suite. Angiographic and perfusion imaging of six patients undergoing pre-operative embolization was performed before surgical resection in a combined X-ray and MR suite. Before and after embolization, selective MRI perfusion was done during dilute contrast injection by MR compatible catheters, positioned in the external and common carotid artery consecutively. Selective injection MR imaging in the angio suite turned out to be an excellent imaging modality for demonstrating treated and untreated tumor areas after embolization [23]. However, selective MRI perfusion in the angio suite might be impractical because of the necessity of using MR-certified catheters and the need to move the patient with the catheter fixed in place from the angiography section to the MR bore. In addition, the requirement of using relatively long acquisition times is accompanied by a higher risk of embolus formation. In addition, only a few hybrid angiographic MRI combinations exist. We encountered no difficulties with the FD CT imaging with selective catheter positions. A precise 3D delineation of the tumor vascular supply was available on-the-spot, rendering a very precise appreciation for the usefulness of pre-operative embolization in consultation with the operating neurosurgeon. The 3D FD CT images in patient no. 2 enhanced the decision to perform the limited tumor embolization, despite the rather low dural tumor supply compared to total tumor volume. Based on our experiences, angiographic and 3D perfusion mapping of final residual tumor supply appeared to be a useful tool for the neurosurgeon in planning the optimal tumor resection approach and procedure with the lowest risk for bleeding complications. The performance of FD CTP PBV imaging after embolization provided an excellent overview of devascularized versus non-devascularized tumor areas.

Our data clearly demonstrated a decline in rPBV when pre-embolization rPBV values were compared with post-embolization values. However, this effect was not significant but can most likely be explained by the low patient numbers included in this study. The calculated mean difference in rPBV of 3.33 with an SD of 2.04 corresponds to a large effect of size of 1.63. The power to find an effect of this size with a paired t-test on four observations is 60%, which illustrates that the observed effect is already of considerable size, even with such a small number of observations. This observation is in accordance with the findings of Wen et al. [21]. Interestingly, in patient no. 1, who showed the highest pre-embolization rPBV value with the largest ∆rPBV value, the highest amount of blood loss (500 mL) was reported. We noticed that embolized tumor portions appeared on the post-embolization perfusion images in two patterns, as hypoperfused and as blacked-out patterns, such as bone structures (Figure 4). Probably, the high density of stagnating contrast medium in embolized tumor regions resulted in a normalization issue in calculating the PBV maps, rendering these areas as bony structures on the PBV maps. The reason for the stagnation of the contrast medium might be the leakage of intravascular contrast into the meningioma, possibly facilitated by the embolization procedure. It is described in magnetic resonance perfusion imaging that meningiomas demonstrate typical time-intensity curves with a rapid signal drop during the first pass of contrast and slow signal return, indicating increased contrast leakage and permeability [24]. This phenomenon was not described in the few published papers about PBV imaging in hypervascular intracranial tumors. The presence of an intratumoral contrast medium after embolization might pose difficulties in calculating the post-embolization PBV scans. Next to a “blacked-out” phenomenon, this may lead to an overestimation of the PBV values on post-embolization scans. This might be a reason for the low ∆rPBV finding in patient no. 2. Future research should further clarify the impact of intratumoral contrast leakage on pre and post-embolization PBV imaging in the angiography suite.

This study had some limitations. First, our sample size is very low, enabling no robust statistical analysis of the collected parameters, allowing only for the appreciation of trends. Within this small group of patients, we did not look into the effect of individual patient-specific parameters on PBV imaging, such as cardiac output, weight, age or anatomical variation of the aortic arch. Further, we assumed an adequate steady-state contrast filling of the tumor after 9 s in aortic root injection and 2 s after selective injection. However, we observed contrast filling of the dural sinuses at the first fill run images, thus, a complete steady-state contrast filling might not be the case. The applied scan timing thresholds and used contrast doses were based on our own experiences with PBV imaging and FD CT imaging in general, in other neurovascular patients. The timing seems to be adequate; however, it has not been validated in other studies. Therefore, additional optimization of the scan protocol should be undertaken, regarding the scan timing and contrast dose delivery.

In our experience, both in the study and also in our clinical practice, the FD CTP PBV imaging is susceptible to motion artifacts. This can be solved by performing the procedures with patients under general anesthesia or by developing angiographic systems using faster C-arm rotation times. Moreover, the implementation of automated calculations in post-processing the scans instead of manual handling might improve the workflow in clinical practice as well.

It is beyond question, however, that this imaging technique requires further exploration. The usefulness of this diagnostic technique needs to be defined more clearly in the future. Nonetheless, conceptually, it helps provide the surgeon insights into (1) the vascularity of the lesion, (2) the main vascular supply to the lesion, and (3) evaluation and monitoring of the effect of pre-operative embolization. By using routine diagnostic imaging studies, the evaluation of quality and quantity of pre-operative embolization might be lacking, resulting in some patients may end up with unnecessary pre-operative embolization. On the other side, denial of pre-operative embolization in some patients, based on conventional pre-operative angiography, might turn out to be a wrong decision during surgery. We hope that research into this new modality will increase our knowledge on the topic of vascularity of the lesion and subsequent embolization.

## 5. Conclusions

Both selective and non-selective FD CTP PBV imaging in patients with meningiomas is, in our experience, an easy to perform procedure in the angio suite. Selective FD CTP provides a robust and convenient insight into perfused volumes and perfusion values before and after pre-operative tumor embolization. The on-the-spot use of the technique in the angio suite enables a more efficient workflow for patients and physicians in vascular mapping for potential pre-operative embolization and for planning tumor resection strategies. Selective quantification of tumor rPBV pre- and post-embolization with the use of ∆rPBV might be used for monitoring the result of performed embolization procedures, especially at the level of tumor parts of special interest to the surgeon, regarding potential bleeding complications.

However, several issues remain unresolved. For instance, the effect of intralesional contrast leakage to FD CT PBV values is unclear. Additional research is warranted to further specify the technique’s role in the setting of pre-operative tumor embolization, hopefully leading to a better balancing of the potential high embolization risks versus the intended benefits for subsequent surgery. Finally, FD CTP imaging might be applied in other interventional neuroradiology indications as well and should be further explored in order to raise awareness about the technique. For example, PBV imaging after both selective and non-selective contrast injection can play a role in the treatment of large vessel acute stroke, in the treatment of delayed cerebral ischemia after subarachnoid hemorrhage, or in the evaluation of patients after extracranial to intracranial bypass surgery.

## Figures and Tables

**Figure 1 diagnostics-12-01185-f001:**
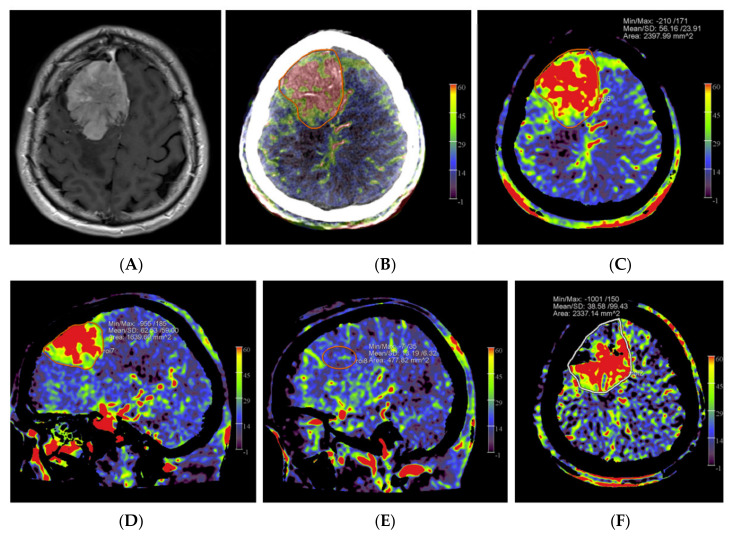
Example of PBV images pre- and post-embolization with manually drawn regions of interest (ROIs) in a patient with a right frontal convexity atypical meningioma (patient no. 2). Axial post-contrast T1-weighted MRI image (**A**) shows the tumor with vivid enhancement and intralesional large vessels in a spoke wheel pattern. According to Wen et al. [21], handheld ROIs for PBV value measurements are drawn on PBV images after aortic root contrast injection in axial plane and in sagittal plane before embolization (axial reformation in (**C**) and sagittal reformation in (**D**) and after embolization (axial reformation in plane (**F**), reformation in sagittal plane is not shown) at the level of the largest tumor area. To confirm correct delineation of the PBV measurement area, PBV series were fused with the fill run series (**B**). Reference PBV measurement on the contralateral side was done using an elliptical ROI in sagittal plane reformation (**E**).

**Figure 2 diagnostics-12-01185-f002:**
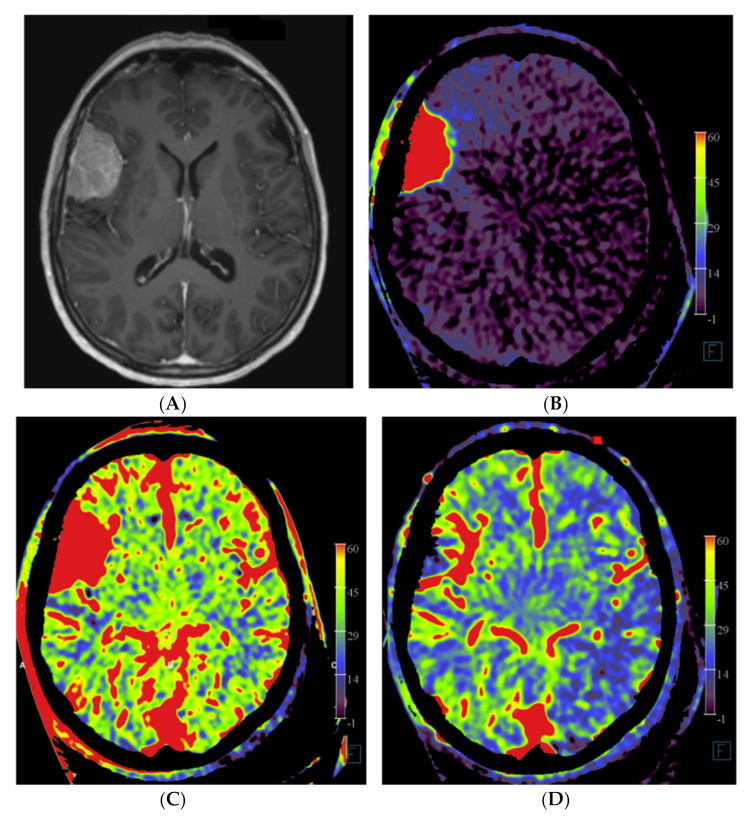
Imaging of a patient with a right-sided perisylvian meningioma. (**A**) Axial post-contrast T1-weighted MRI image of patient no. 4 shows the presence of an enhancing, extra-axially located tumor in the right perisylvian area. (**B**) Pre-embolization axial PBV reformation obtained after selective right external carotid artery injection and (**C**) after aortic root contrast injection shows a hypervascular tumor. In consultation with the operating neurosurgeon, a pre-operative embolization was performed. (**D**) Post-embolization axial reformation of whole brain PBV scan shows devascularization of the portion supplied by the dural feeder.

**Figure 3 diagnostics-12-01185-f003:**
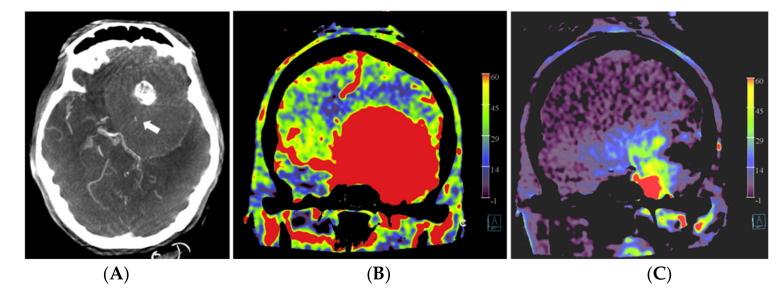
Patient no. 3 presented with a very large sphenoid ridge meningioma on the left side. (**A**) Axial 3 mm reformation based on fill run images shows the large tumor with encasement of the distal internal carotid artery (arrow) and the middle cerebral artery on the left. (**B**) Coronal reformation of whole brain PBV perfusion imaging compared to (**C**) coronal reformation of selective PBV perfusion by injection of the left external carotid artery shows predominant pial blood supply and only limited dural supply to the tumor. In this case, pre-operative embolization was not pursued because it was considered not helpful for reducing the surgery risk in comparison to the risk of the embolization procedure.

**Figure 4 diagnostics-12-01185-f004:**
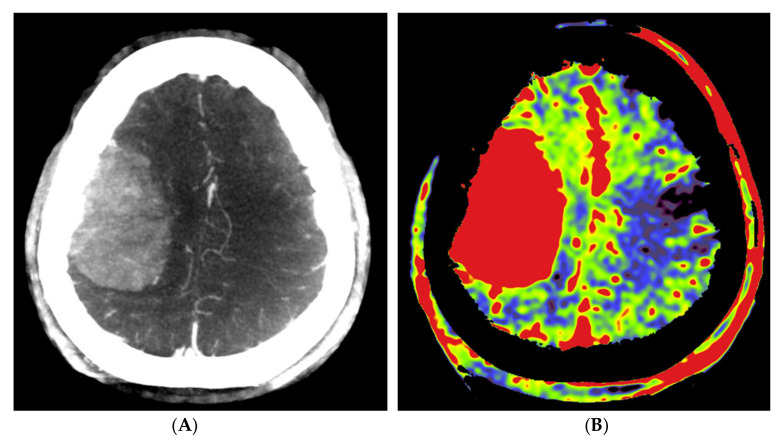
Pre- and post-embolization fill run and PBV mapping of a patient (no. 1) with a large right hemispheric meningioma. (**A**) Pre-embolization axial reformations of fill run and (**B**) whole brain PBV perfusion show dense, homogeneous, hypervascular enhancement of the tumor. (**C**) Post-embolization axial fill run reformation demonstrated tumoral areas with decreased enhancement and large areas with pooled contrast medium. (**D**) The corresponding axial whole brain PBV reformation shows both devascularized (arrows) and blacked-out (*) areas with pooled contrast medium.

**Table 1 diagnostics-12-01185-t001:** This table summarizes the clinical characteristics of the tumor for the five patients.

Patient No	Diagnosis	Tumor Location	Tumor Size (cm)	Arterial Supply	Operation Time (hours)	Blood Loss (mL)
1	Fibrous meningioma, WHO grade I	Right frontal convexity	6.7 × 4.5 × 4.1	Bilateral ECA + right ICA pial	5	500
2	Atypical meningioma, WHO grade II	Right frontal convexity	5.0 × 3.6 × 4.8	Bilateral ECA + ICA pial + dural	4	N
3 *	Transitional meningioma, WHO grade I	Left sphenoid ridge	7.6 × 6.1 × 4.0	Left ICA pial + dural, limited left ECA	14 (including STA-MCA bypass)	300
4	Meningiothelial meningioma, WHO grade I	Right perisylvian	4.5 × 2.7 × 3.9	Left ECA	8	N
5	Atypical meningioma, WHO grade II	Interhemispheric	3.3 × 2.3 × 3.0	Bilateral ECA + left ACA pial + left VEA dural	3.7	N

WHO = World Health Organization, ECA = external carotid artery, ICA = internal carotid artery, STA-MCA bypass = superficial temporal artery-middle cerebral artery bypass, ACA = anterior cerebral artery, VEA = vertebral artery, N = negligible, blood loss was not measured if <50 mL. * Patient no.3 underwent no embolization.

**Table 2 diagnostics-12-01185-t002:** An overview of tumor volume (mL) pre- and post-embolization, tumor volume (mL) supplied by dural arteries from the external carotid artery (ECA) and rPBV values with ∆rPBV is given.

Patient No	Tumor V in mL (Pre/Post-Embolization)		V Total ECA Dural Vascular Supply in mL (% of Total Tumor V Pre)	rPBV Tumor (mL/1000 mL)		∆rPBV
	PRE	POST		PRE	POST	
1	75	40	34 (45%)	8	2.4	5.6
2	56	42	18 (32%)	5.4	4.7	0.7
3 *	193	-	61 (31.5%)	5.4	-	-
4	35	2	32 (91%)	4.5	0.6	3.9
5	20	8	14 (70%)	5.1	2	3.1

V = volume; rPBV: relative parenchymal blood volume values calculated according to Wen et al. [21]. ∆rPBV = rPBVpre-embolization-rPBVpost-embolization. * Patient no.3 underwent no embolization.

## Data Availability

Not applicable.

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
