# Peer review of "Selective Angiographic Flat Detector Computer Tomography Blood Volume Imaging in Pre-Operative Vascular Mapping and Embolization of Hypervascular Intracranial Tumors—Preliminary Clinical Experience"

_diagnostics, 2022, doi:10.3390/diagnostics12051185_

Round 1
Reviewer 1 Report
The paper deals with the pre-operative embolization of hypervascular intracranial tumors to reduce bleeding complications during resection. The subject is timely and interesting. Minor language and editing are needed. However the number of patients is very little and this should be improved recruiting more patients in order to have more data to be considered.
In our opinion, the paper should be re-submitted after the recruitment of more patients and a new evaluation of data.
Author Response
Comments:
The paper deals with the pre-operative embolization of hypervascular intracranial tumors to reduce bleeding complications during resection. The subject is timely and interesting. Minor language and editing are needed. However the number of patients is very little and this should be improved recruiting more patients in order to have more data to be considered.
In our opinion, the paper should be re-submitted after the recruitment of more patients and a new evaluation of data."
Response:
We agree that we have only few patients in our series. Unfortunately, we do not have the opportunity to include many more patients in the upcoming period prospectively. However, in consultation with a clinical statistician only one more extra patient included would lead to a statistical significant sample size. This would be possible to realize the next weeks to months. Would you agree to accept the manuscript if we include one or two more patients?
Reviewer 2 Report
"This paper explored a new imaging technique to quantify tumor rPBV pre- and post embolization, for monitoring the result of performed embolization procedures. 5 patients were included for the experiments. Qualitative results look promising. A few concerns are raised for the consideration,
1) The rPBV maps visually showed the decreased blood volume after embolization. I am wondering whether there is any standard reference to quantitatively evaluate the validity of the maps?
Response to comment 1:
This is a very justified comment. We had our thoughts about it as well.
Regarding the quantitative measurements, in principle, in our opinion, the only true references for quantitative perfusion parametric map validation are based on invasive methods. However, as less invasive alternatives, you can use validated MRP and CTP for references (Stille et al published in Neuroradiology a systematic review about diagnostic accuracy of FP CT comparing with CTP and MRP in 2019, doi: 10.1007/s00234-019-02285-y). But there are drawbacks. For instance, even if you have MDCT and MRI modalities available in the angiosuite, you still have uncertainty about the exact equal physiological circumstances between angiographic PBV imaging and following conventional PBV mapping within seconds to minutes. Besides, it is well known that quantification of perfusion maps with MDCT perfusion can vary considerably between vendors, even using the same source images. This will be most likely the case for angiographic PBV as well. Therefore, we opted for a semi-quantitative method in the form of rPBV measurements, as currently used in clinical practice for MRP and CTP as well.
2) As mentioned in the paper, the testing sample size is too small. More comprehensive validations are required.
Response to comment 2:
We agree that we have only few patients in our series. Unfortunately, we do not have the opportunity to include many more patients in the upcoming period prospectively. However, in consultation with a clinical statistician only one more extra patient included would lead to a statistical significant sample size. This would be possible to realize the next weeks to months. Would you agree to accept the manuscript if we include one or two more patients?
3) There are any suggestions to improve the imaging technique itself?
Response to comment 3:
- We have the impression that our used contrast concentrations are on the high side. We have performed test runs with different contrast doses on a tube model and we observed a plateau of the PBV measurements after a certain amount of contrast. However, due to methodological uniformity we performed the exams with the given dose concentrations (based on the vendor recommendations and our own previous experiences in other neurovascular cases). In the future, we plan to optimize our scanning protocols.
- The technique is susceptible to motion artifacts (on pbv mapping more pronounced than the fill runs). The use of general anesthesia and faster rotation times might overcome this problem.
- PBV imaging of the posterior fossa proofs to be more prone to beam hardening artifacts.
- Post processing might be further optimized by implementing automated calculations, instead doing it manually
4) what it the road bock to translate it to the clinical practice?"
Response to comment 4:
Following potential reasons might limit the technique in general clinical practice:
- The technique is not widely known with only a few publications, therefore “unknown is unloved”. We hope this will improve in the future with ongoing use of the technique and additional publications about the technique
- Adequate acquisition of the technique is not that difficult. However, post processing and analysis of the technique takes some time and experience.
- As far as we know, the technique is developed by Siemens company for the angiography suite. It is a software application. For future use, the software should also be made available by other vendors. At the moment, the technique is under investigation for application in limited neurovascular indications. Very publications exist about its use in the liver. However, future use for other vascular embolization should be investigated.
Round 2
Reviewer 1 Report
all tasks were assessed
Author Response
- We adjusted the title as suggested into " Selective angiographic flat detector computer tomography blood volume imaging in pre-operative vascular mapping and embolization of hypervascular intracranial tumors - preliminary clinical experience".
- We incorporated minor adjustments in the discussion and conclusion sections of the manuscript based on the comments 3 and of 4 of reviewer 2.
- Few minor corrections were made regarding spaces and punctuation marks (dots, brackets, quote marks).
Reviewer 2 Report
Thank for the responses to my comments. As mentioned by the authors, it would be better to have a statistical significant sample size, before publishing this paper. Otherwise, I do not have other comments.
Author Response

(The authors gave the same response as above.)
